🔓 | **Open Peer Review** | Virology | Methods and Protocols

# Evaluating the susceptibility of various common cell lines and assessing inactivation conditions to Mpox virus

Shu-Chen Hsu,[1,2] Ping-Cheng Liu,[2,3,4] Shan-Ko Tsai,[1,2] An-Yu Chen,[1,2] Hui-Ping Tsai,[2] Jun-Ren Sun,[5,6,7] Ti-Yu Li,[8] Pei-Yu Hsieh,[8] Jyh-Yuan Yang,[8] Tein-Yao Chang[2,7]

**ABSTRACT** Over the past decade, numerous infectious diseases have emerged. In 2022, the World Health Organization declared the Mpox outbreak a global public health emergency, as the virus spread rapidly to over 107 countries. As of May 2023, Taiwan continued to observe sporadic instances of the disease. Understanding host cell susceptibility and viral inactivation conditions is crucial for elucidating viral mechanisms and developing effective antiviral therapies or vaccines. In this study, we evaluated the susceptibility of commonly utilized laboratory cell lines to the Mpox virus by serial passage and assessed various conditions for virus inactivation, such as heat treatment or reacting with multiple reagents. Our results revealed that the Mpox virus could infect multiple cell lines, with the BSC-1 cell line being the most susceptible. Heating at 56°C for 10 min or longer rendered the virus non-infectious, indicating its thermosensitivity. Furthermore, widely used reagents, such as TRIzol, alcohol, Micro-Chem Plus, bleach, and formalin, completely inactivated the virus at recommended concentrations. However, radioimmunoprecipitation assay buffer without SDS should be used with caution, as it may not fully eliminate infectious particles. Our results provide pivotal reference data for future studies and standardization efforts in Mpox virus research, enhancing our understanding and management of this emerging pathogen.

**IMPORTANCE** The global resurgence of Mpox highlights the urgent need for robust diagnostic, therapeutic, and biosafety strategies. However, critical gaps remain regarding its replication across various cell types and the effectiveness of disinfection methods. This study systematically evaluates the susceptibility of commonly used laboratory and clinical cell lines to Mpox virus, providing key insights to optimize viral isolation in research and diagnostic settings. Additionally, by assessing the effectiveness of standard disinfectants against Mpox, this work strengthens biosafety protocols for healthcare and high-containment laboratories. These findings have direct implications for public health preparedness, guiding both laboratory practices and biosafety measures.

**KEYWORDS** Monkeypox, Mpox, cell lines susceptibilities, disinfectants, thermosensitive

The World Health Organization declared the Mpox (Monkeypox) outbreak in Europe in May 2022 a global public health emergency. The outbreak was first identified in the United Kingdom and rapidly spread to over a hundred countries within less than 6 months (1). The first imported case of Mpox was reported in July 2022 in Taiwan. To date, over 107 countries have reported Mpox outbreaks, with more than 87,000 confirmed cases at the time of writing this article (2).

The Mpox, cowpox, and variola viruses belong to the Orthopoxvirus genus within the *Poxviridae* family. Compared to other viruses, the Mpox virus has a large size and a linear, double-stranded DNA genome of approximately 197 kbp (3). The virus is classified into two clades: clade I (Congo Basin) has a mortality rate approaching 10%. In contrast,

**Peer Reviewer** Chee-Keng Mok, Changping National Laboratory, Beijing, China

Address correspondence to Tein-Yao Chang, teinyao@gmail.com.

The authors declare no conflict of interest.

clade IIa (West Africa) shares over 95% genomic similarity with clade I but has a mild mortality rate of less than 1% (4, 5). A comparison of clade I and clade II reveals that although most of their genomes exhibit similarity, clade II demonstrates variations attributed to mutations or deletions in several virulence-associated genes, such as D10L, B10R, or B14R. As a result, clade II Mpox exhibits a lower mortality rate than clade I (6). African animal importation caused a localized outbreak of Mpox, with a lineage belonging to clade IIa, in the United States in 2003 (7). However, the 2022 Mpox outbreak was classified as clade IIb because of its similarity to the clade IIa genome sequences, despite its lower mortality rate (6–9). Clades I and IIa cause zoonotic diseases, whereas clade IIb shows broad human-to-human transmission. The respiratory transmission of the Mpox virus was inefficient in the laboratory, with direct animal contact and close human interactions critically regulating human transmission (10–12). The incubation period for the disease ranges from 3 to 17 days. Individuals with Mpox may present with a characteristic rash that can manifest on various body parts, including the hands, feet, chest, face, mouth, or near the genital region (13). Statistical analysis indicated that the $R_0$, which denotes the number of a single infected individual that can cause spread to another, of the 2022 Mpox outbreak was estimated to vary from 1.56 to 3.80 by country (14).

Understanding cell susceptibility to viral infections and identifying effective inactivation conditions for emerging pathogens are important in virology research and public health. The former provides valuable insights into viral pathogenesis, host-virus interactions, and the development of antiviral strategies (14–17). Nevertheless, most previous studies on the Mpox virus chose cell lines based on references to other poxviruses (18), without conducting specific assessments of the susceptibility of conventional laboratory cell lines to the Mpox virus. This lacuna in the evaluation may have introduced bias in choosing particular cell lines. Establishing optimal inactivation conditions is crucial for ensuring laboratory safety, preventing viral transmission, and preserving sample integrity.

In this study, we used various commonly employed laboratory cell lines to evaluate their susceptibility to the Mpox virus. In addition, we examined the ability of disinfectants and cell lysis reagents widely used in research environments and experimental settings to inactivate viruses effectively. Our findings establish a comprehensive set of experimental conditions that provide valuable reference data for future studies and standardization efforts on the Mpox virus. Moreover, the findings from this study would help determine the effectiveness of different disinfectants in inactivating the Mpox virus for controlling the global epidemic.

## MATERIALS AND METHODS

### Cells and virus

Cell lines from the American Type Culture Collection, including BSC-1, HeLa, LLC-MK2, MRC-5, RD, Vero E6, BHK-21, HEp-2, Huh7, MDCK, CHO, HEK-293, and MM55.K, were cultured in Dulbecco's Modified Eagle Medium (DMEM; Cat# SH30243.02, HyClone, Logan, USA), supplemented with 10% fetal bovine serum (FBS; Cat# 26140079, Gibco, Grand Island, USA) and 1% antibiotic-antimycotic (Cat# 15240-062, Gibco, Grand Island, USA) at 37°C with 5% $CO_2$. The Mpox virus strain, hMpxv/Taiwan/CVDCDC-110-231642/2022 (GISAID Accession ID: EPI_ISL_13632071), was isolated from a Taiwanese patient, a man in his 20s with a documented history of travel to Germany. The BSC-1 cells were used to propagate the virus. Following adsorption, the inoculum was removed and replaced with DMEM supplemented with 2% FBS (E-2 medium). A viral cytopathic effect (CPE) was observed, and virus harvesting involved subjecting the cells to three freeze-thaw cycles and collecting the supernatant and cell debris. This study was approved by our Biosafety Committee (reference number B1-112-0006). All live-virus experiments were conducted in a Biosafety Level 4 (BSL-4) laboratory.

## Different cell lines' susceptibility test

To evaluate Mpox virus susceptibility, each cell line was seeded at $4 \times 10^5$ cells per well in 12-well plates (Cat# PC312-0050, GeneDireX, New Taipei City, Taiwan) in duplicate the night before the experiment. Upon removal of the growth medium, the cells were infected with 300 µL of Mpox at a multiplicity of infection (MOI) of 0.001. Following an hour of incubation for viral adsorption at 37°C, the virus was discarded and replaced with 2 mL/well of E-2 medium. Incubating these cells at 37°C in 5% $CO_2$ for 4 days was the first passage of infection. Next, we placed the 12-well plate in a −80°C freezer overnight to ensure complete freezing, then transferred it to a 37°C incubator for 1 h to thaw. This constituted one freeze-thaw cycle. Based on previous studies, three freeze-thaw cycles were performed to lyse the cells and ensure the complete release of viral particles. The supernatant was mixed with cell debris, and 300 µL was inoculated into fresh cells. Following an hour of virus adsorption, we replaced the culture medium with E-2 medium and incubated the cells at 37°C in 5% $CO_2$ for 4 days, which we define as the second passage (P-2) (15, 19). All samples were collected and stored at −80°C until real-time quantitative PCR analysis or plaque assay.

## Real-time qPCR

The Mpox virus G2R gene copies were quantified using qPCR with specific primers. The specific primers used were G2R-G_F (5′-GGAAAATGTAAAGACAACGAATACAG-3′), G2R-G_R (5′-GCTATCACATAATCTGGAAGCGTA-3′), and G2R_G_probe (5′-FAM-AAGCCGTA ATCTATGTTGTCTATCGTGTCC-3′-BHQ1) (20). All primers and probes were purchased from Integrated DNA Technologies. The total nucleic acids of the samples were extracted using a TANBead Maelstrom 4800 automatic extraction system with a TANBead OptiPure Viral kit (Cat# M665S46, TANBead, Taoyuan, Taiwan). Subsequently, the extracted nucleic acids were analyzed using the Roche LC480 system (Roche, Basel, Switzerland) with automatic threshold and baseline settings. The QuantiNova Probe RT-PCR Kit (Cat# 208252, QIAGEN, Hilden, Germany) was used as the reagent, and the user's manuscript was followed to perform the experiments.

## Plaque assay

For the plaque assay, BSC-1 cells were seeded in 24-well plates (Cat# 3524, Corning, NY, USA) at a density of $8 \times 10^4$ cells per well and incubated at 37°C in 5% $CO_2$ overnight. The virus samples were serially diluted 10-fold by E-2 medium, and 200 µL of the dilutions were added to the cells. The cells were incubated for 1 h at 37°C with shaking at 15 min intervals to facilitate viral adsorption. Next, the virus was removed, and 1 mL of E-2 medium was added to the cells, which were incubated for 3 days at 37°C with 5% $CO_2$. The supernatant was discarded, and the monolayer was fixed with 1 mL of 10% formalin (Cat# HT501320, Sigma-Aldrich, St. Louis, USA) for at least 1 h, at room temperature. The monolayer was stained with 0.4% crystal violet (Cat# C0775, Sigma-Aldrich, St. Louis, USA) in 20% methanol (Cat# 1.06009.1000, Merck, Darmstadt, Germany) and washed with tap water to visualize plaques. Plaques were quantified and recorded as PFU per milliliter.

## The median tissue culture infectious dose

The assay was performed to evaluate viral infectivity. Briefly, BSC-1 cells at a concentration of $8 \times 10^3$ cells/well were seeded in a 96-well plate (Cat# PC396-0100, GeneDireX, New Taipei City, Taiwan) and cultured overnight. Viral supernatants containing cell debris were subjected to 10-fold serial dilutions in E-2 medium. BSC-1 cells were washed twice with 200 µL of phosphate-buffered saline (PBS) per well. Subsequently, 100 µL of the diluted samples was inoculated into the 96-well plates pre-seeded with BSC-1 cells. Each dilution was performed in sextuplicate. Following incubation at 37°C with 5% $CO_2$ for 5 days to allow CPE development, the plates were fixed with 200 µL of 10% formalin and subsequently stained with crystal violet to visualize the CPE.

## Viral growing curve

To characterize Mpox virus replication kinetics in various cell lines, a multi-step growth-curve analysis was performed. A total of $8 \times 10^4$ cells per well were seeded into 24-well plates and incubated overnight. Cells were then infected with Mpox virus at an MOI of 0.001. After 1 h of adsorption at 37°C, the inoculum was removed and replaced with 500 µL of fresh E-2 medium. Cultures were maintained at 37°C, and samples were collected at the indicated time points. Each sample underwent three freeze-thaw cycles to release intracellular virus, after which viral titers were quantified by qPCR and plaque assay. All conditions were tested in quadruplicate.

## Plaque assay conditions test

Two cell lines, BSC-1 and Vero E6, were evaluated in overlapping media to compare the optimized plaque assay conditions. Briefly, $8 \times 10^4$ cells were seeded into each well of a 24-well plate and incubated overnight. Virus samples were then serially diluted and added to the cells at a volume of 200 µL, followed by a 1 h incubation at 37°C to allow virus adsorption. The plate was gently rocked every 15 min during adsorption. After removing the inoculum, cells were overlaid with 1 mL of either methylcellulose (Cat# M7140-1KG, Sigma-Aldrich, St. Louis, USA) in E-2 medium at concentrations of 1.55%, 0.78%, and 0.4%, or microcrystalline cellulose (Avicel RC-591, DuPont, Wilmington, USA) in E-2 medium at concentrations of 1.25% and 0.63%. The control group was overlaid with E-2 medium only. Each group performed in quadruplicate. The plate was then incubated at 37°C with 5% $CO_2$ for 3–6 days. After fixation with 10% formalin, plaques were visualized by staining with crystal violet, and the experimental results were recorded.

## Viral inactivation

For the heat-inactivation assay, 100 µL of Mpox virus ($8 \times 10^4$ PFU) was dispensed into eight-strip tubes (Cat# MB-P08-A, Gunster, Taiwan). Samples were divided into the following groups: negative control, unheated control, and heat treatments at 56°C, 65°C, and 95°C. All groups were processed in octuplicate. Heating was performed in a PCR thermocycler (TCLT9610, Blue-Ray Biotech, Taiwan) to ensure precise temperature and experimental consistency. The samples were harvested at the time points of 1, 3, 5, 10, 15, and 30 min. A total of 100 µL of E-2 medium was added to each tube to obtain a final volume of 200 µL. The entire volume was inoculated onto BSC-1 cells seeded in 24-well plates and allowed to adsorb for 1 h at 37°C. Following adsorption, the viral supernatant was removed, fresh E-2 medium was added, and the cultures were incubated for 3 days. Finally, cells were fixed with 10% formalin and stained with crystal violet to visualize plaque formation (21).

For evaluating the effect of lysis buffers and disinfectants, $8 \times 10^4$ PFU of Mpox virus was mixed with the respective solutions listed in Table 3 and incubated for 10 min at room temperature. The reaction volume was then adjusted to 4 mL with PBS and passed through an Amicon Ultra-4 100 kDa centrifugal filter (Millipore, Ireland) at 3,500 rpm by centrifuge (DSC-N158A, Digisystem Laboratory Instrument, Taiwan) until ~250 µL remained. The retentate was brought back to 4 mL with PBS, and the wash was repeated twice more to remove residual chemicals. The retentate remaining on the filter was brought to a final volume of 500 µL with E-2 medium. Each experimental condition was tested in quadruplicate. Then, the viruses were inoculated into BSC-1 cells in a 24-well plate and incubated for 3 days. The cells were fixed and stained with crystal violet to visualize the viral plaques.

To validate the effect of formalin and paraformaldehyde on inactivating the virus. BSC-1 cells were grown to full confluence in 24-well plates. After PBS washes twice, the cells were infected with the virus at an MOI of 0.1 and incubated for 24 h. The medium was then removed, and 1 mL of a chosen fixative was added to each well: 10%, 5%, or 1% formalin, or 4% or 1% paraformaldehyde. Mock-infected controls were subjected to

the same fixation procedures in parallel. Viral-control wells received 1 mL PBS instead of fixative. All conditions were set up in four replicates. Fixation lasted 1 h at room temperature. The fixatives were discarded, and the monolayers were rinsed three times with 1 mL PBS to remove chemical residues. Each well was then loaded with 500 µL of PBS, and the plate underwent three freeze-thaw cycles to release virus from the cells. The whole suspension was collected, and 200 µL of it was used to infect fresh confluent BSC-1 monolayers. After 1 h of adsorption, the inoculum was removed, 0.5 mL E-2 medium was added, and the cultures were incubated for 72 h. Finally, cells were fixed with 10% formalin for 1 h and stained with crystal violet to visualize the plaques.

## Viral freeze-thaw cycles stability test

To evaluate Mpox virus stability, 100 µL aliquots containing $8 \times 10^4$ PFU were dispensed into microcentrifuge tubes (Cat# 72.692.005, Sarstedt, Nümbrecht, Germany) and subjected to one to six successive freeze-thaw cycles; each condition was tested in triplicate. After the designated cycles, viral titers were quantified by plaque assay and qPCR as described above.

## Statistical analysis

The number of independent replicates for each experiment is indicated in the corresponding Materials and Methods subsections. Data are expressed as mean ± SD. Descriptive statistics and hypothesis testing were performed in GraphPad Prism 8 (GraphPad Software, San Diego, CA, USA) or Microsoft Excel 2019. Differences between the two groups were evaluated with Student's $t$-test, and $P < 0.05$ was considered statistically significant.

## RESULTS

### The Mpox virus propagates in various conventional cell lines

The P1-0 h represents the time point immediately after 1 h of viral absorption, serving as a baseline indicating the initial viral load for Mpox virus replication in various cell lines, ranging from $3.84 \times 10^2$ to $3.68 \times 10^4$ copies/mL (Fig. 1A). All cells showed viral titers ranging from $8.49 \times 10^3$ to $1.52 \times 10^7$ copies/mL during P1. The fold change in viral replication was calculated for each cell type, yielding log-transformed values from 0.98 to 4.06 (Fig. 1A; Table 1). The fold-change analysis classified the cells into four groups based on their viral replication capacity. Replication fold changes exceeding 1,000-fold were observed in BSC-1, MRC-5, Vero E6, and 293T cell lines, whereas MK-2, RD, and MM55.K cells exhibited replication fold changes above 100-fold. BHK-21, MDCK, and CHO cells demonstrated a replication fold change above 10-fold, and Hep-2 and Huh-7 cells had the least of less than 10-fold (Table 1). In contrast, during the viral cultures' P-2, viral titers ranged from $4.43 \times 10^3$ to $2.72 \times 10^7$ copies/mL (Fig. 1A). BHK-21 and Hep-2 cells showed increased viral replication, while MDCK cells exhibited decreased replication (Table 1).

Six cell lines were selected to analyze the Mpox virus's growth curve. After virus adsorption at an MOI of 0.001, the virus and cell debris were collected at various time points, and CPE was observed. The results categorized the cell lines into three groups. Vero E6, MRC-5, RD, and BHK-21 cells exhibited viral replication from 6 to 24 h, reaching a peak at 48 h ($\sim 10^5$ PFU/mL). BSC-1 cells showed rapid viral replication, starting at 24 h and reaching a peak at 72 h ($\sim 10^{6.8}$ PFU/mL). MM55.K cells exhibited slow viral replication, with viral levels plateauing at 72 h (Fig. 1B). Similar results were observed for viral genome copies. The highest viral copy number was observed in BSC-1 cells at 96 h post-infection. However, viral replication was observed as early as 6 h post-infection and peaked at 48 h in most cell lines, except BHK-21 and MM55.K, with delayed viral growth at 24 h (Fig. 1C). CPEs were observed in BSC-1, RD, BHK-21, and MRC-5 cells 48 h post-infection, whereas Vero E6 and MM55.K cells showed CPEs 72 h post-infection (Fig. 1D).

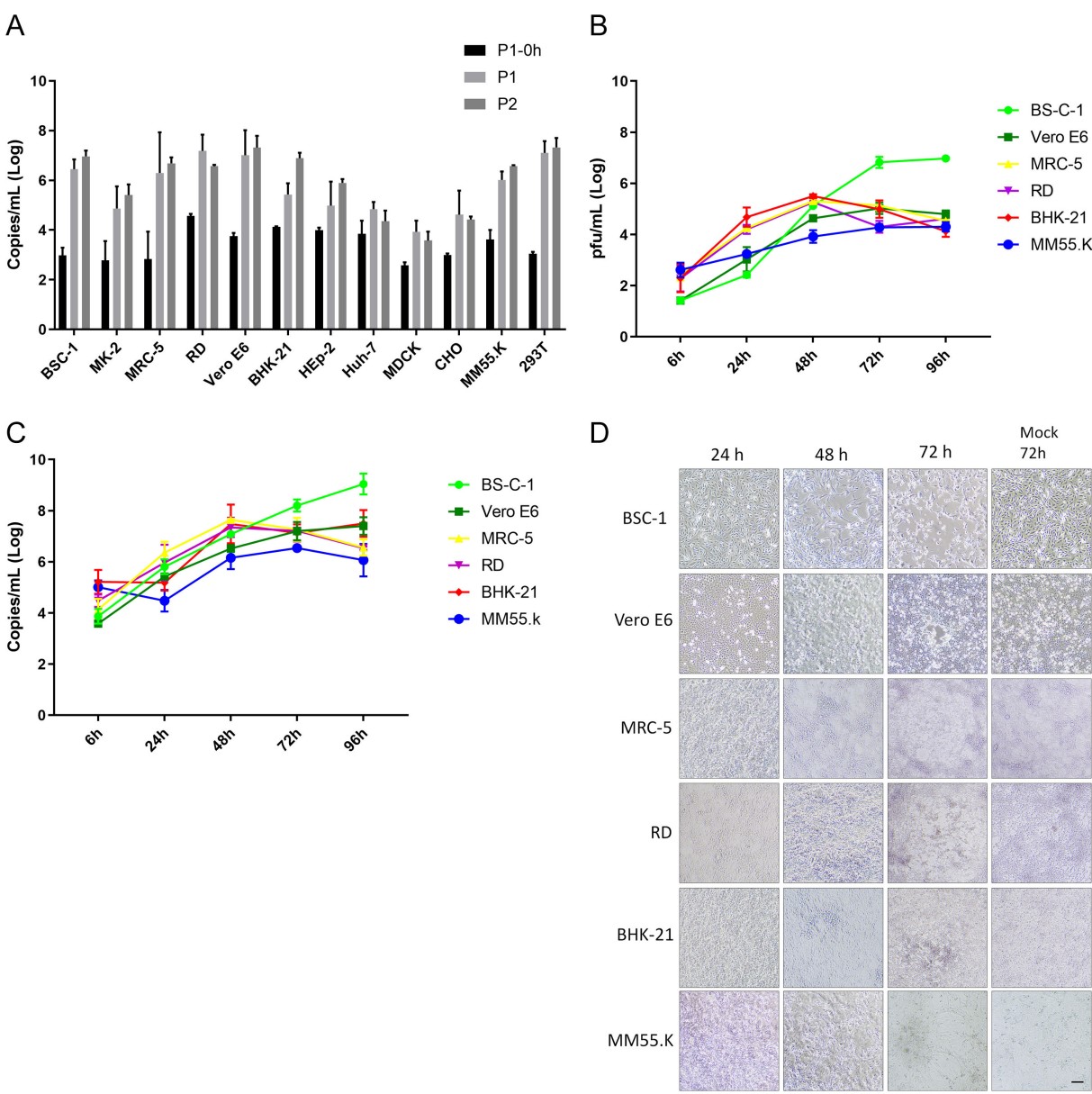

**FIG 1** The susceptibility of different cell lines to Mpox virus and viral growth curve. (A) The viral load in different cell lines that were infected with the Mpox virus for 96 h. The P1-0 h represents the initial viral titers after adsorption. P1 and P2 mean serial passage of the Mpox virus in various cell lines. The viral growth curve was determined in different cell lines by inoculating cells with an MOI of 0.001 and harvesting them at various time points. The viral load was quantified by qPCR (B) and plaque assay (C). (D) CPEs of Mpox virus were observed in various cell lines. Scale indicates 100 µm.

Our results indicated that the Mpox virus could propagate in various cell types, and within 48 h of post-infection, most cells exhibited similar viral titers. However, we chose BSC-1 cells for viral cultivation to achieve higher viral yields.

## The Mpox virus efficiently forms viral plaques in BSC-1 cells

We tested the plaque assay conditions for the Mpox virus using BSC-1 and Vero E6 cell lines and compared the effects of different overlapping media. Mpox virus formed relatively smaller plaques in Vero E6 cells, which were clearly visible to the naked eye on day 6. In contrast, clear plaques were observed as early as day 3 in the BSC-1 cells. The overlapping media used in this experiment did not significantly affect the number of viral plaques, except in the 1.55% methylcellulose group, which produced smaller

TABLE 1 The viral replication fold change

| | Copies/mL fold change (Log) | |
| --- | --- | --- |
| | P1 | P2 |
| BSC-1 | 3.47 | 3.98 |
| MK-2 | 2.09 | 2.63 |
| MRC-5 | 3.46 | 3.85 |
| RD | 2.62 | 2.00 |
| Vero E6 | 3.26 | 3.56 |
| BHK-21 | 1.30 | 2.76 |
| HEp-2 | 0.98 | 1.89 |
| Huh-7 | 0.99 | 0.51 |
| MDCK | 1.34 | 0.99 |
| CHO | 1.63 | 1.43 |
| MM55.K | 2.40 | 2.96 |
| 293T | 4.06 | 4.28 |

plaques than the other groups (Fig. 2A). The viral plaques formed effortlessly, even when the E-2 culture medium was used, reflecting the intercellular spreading characteristic of the virus. Our experimentation with various overlay media formulations demonstrated the consistent efficiency of the Mpox virus plaque formation in BSC-1 cells. However, it is noteworthy that comet-shaped plaques might form when using the E-2 medium alone.

Furthermore, we compared the conditions for titrating the Mpox virus using median tissue culture infectious dose ($TCID_{50}$) in BSC-1 cells and tested the results for 3–6 days (data not shown). The $TCID_{50}$ collected on day 6 demonstrated either the presence or absence of CPEs (Fig. 2B). Additionally, we summarized the viral loads obtained from the plaque assay, $TCID_{50}$, and qPCR for the same specimen tube (Table 2).

## Most reagents and disinfectants effectively inhibited Mpox virus activity at conventional concentrations

Our findings indicate that heat treatment of the Mpox virus at 56°C for 5 min can reduce plaque counts while extending the treatment to 10 min or longer achieves complete viral inactivation (Fig. 3A). Heat treatment is one of the most convenient methods for pathogen inactivation. Current pathogen detection methods rely on nucleic acid detection (such as real-time qPCR) as the primary diagnostic test in the laboratory because of its high sensitivity, accuracy, and reliability (22, 23). Therefore, we further evaluated how heat treatment at different temperatures might damage the viral genome by heat to assess its impact on the sensitivity of nucleic acid testing. This investigation did not yield significantly different Ct values among the virus samples subjected to differential heat treatment. Interestingly, our findings revealed that the average Ct value of the virus treated at 56°C was slightly higher than that of those treated at 65°C and 95°C (Fig. 3B), suggesting that 56°C treatment for inactivation might lead to a lower viral genome copy number (we will further discuss later).

We also assessed the effects of commercial lysis buffers and disinfectants on Mpox virus infectivity. Briefly, $8 \times 10^4$ PFU of Mpox virus was incubated in each reagent for 10 min, residual chemicals were removed with Amicon Ultra-4 100 kDa centrifugal filters, and infectious virus was quantified by plaque assay. Complete loss of plaque formation was considered evidence of inactivation. Under these conditions, most reagents abolished Mpox infectivity at their standard working concentrations (Fig. 4A). The item details and statistics results from multiple repeats are shown in Table 3. Because the chemical resistance of the filter might influence the outcome, we evaluated the compatibility of each formulation. Based on the manufacturer's user guide and safety data sheets, the maximal concentrations of active ingredients that reached the membrane during our wash protocol were below the recommended limits for all reagents except Trizol. To confirm that Trizol did not compromise filter integrity, we

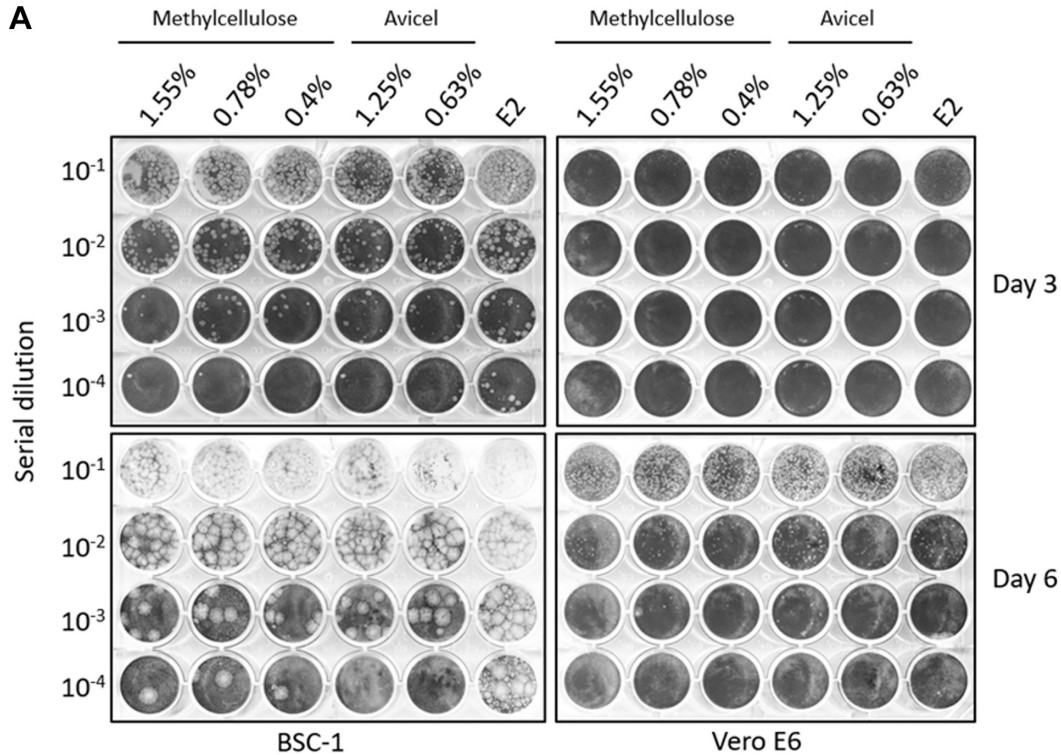

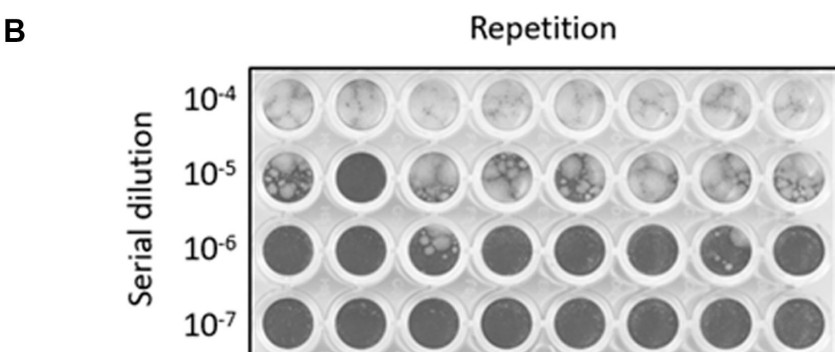

**FIG 2** The conditions test of viral load quantification. (A) Two types of cells, BS-C-1 and Vero E6, were infected with the Mpox virus and overlaid with varying concentrations of Methylcellulose or Avicel (microcrystalline cellulose) as the overlapping medium. The results were collected on day 3 and day 6 post-infection to assess the optimal conditions for the plaque assay. (B) The median tissue culture infectious dose ($TCID_{50}$) was harvested on day 6 post-infection, BS-C-1 inoculating serial dilution of Mpox virus.

performed an additional validation experiment (Fig. S1). The results showed identical virus-retention efficiencies for Trizol-treated and untreated filters with no virus detected in the flow-through, demonstrating that the filter remained fully functional. Unexpectedly, a few viral plaques persisted after repeated treatments with commercial radioimmunoprecipitation assay (RIPA) buffer, indicating that a 10 min exposure was insufficient

**TABLE 2** Comparative analysis of viral quantification across different methods

|  | **Viral loads** |  |
| --- | --- | --- |
| qPCR | $1.12 \times 10^{6} \pm 5.52 \times 10^{5}$ | Copies/mL |
| Plaque assay | $1.98 \times 10^{6} \pm 1.19 \times 10^{5}$ | PFU/mL |
| $TCID_{50}$ | $4.29 \times 10^{5} \pm 5.59 \times 10^{4}$ | $TCID_{50}$/mL |

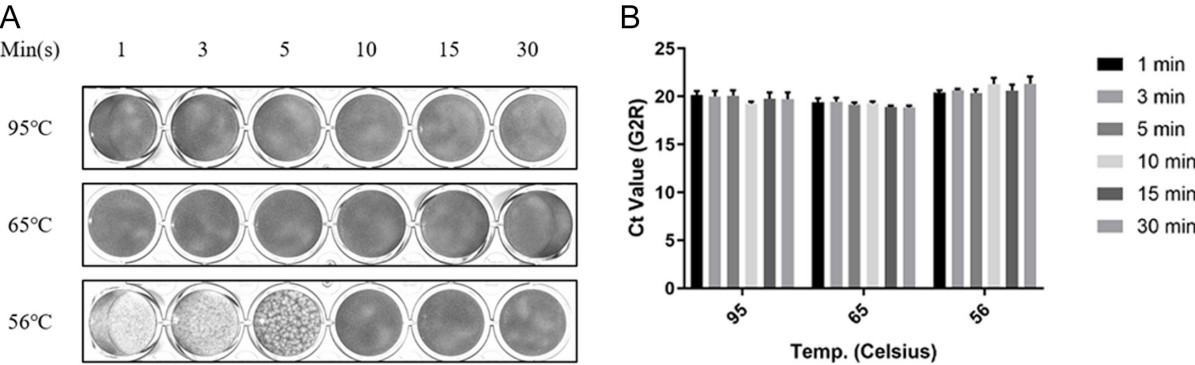

FIG 3 Heat treating the Mpox virus. (A) Mpox virus ($8 \times 10^4$ PFU) was treated at 56°C, 65°C, and 95°C for various time intervals, and plaque formation was observed. (B) The stability of Mpox virus nucleic acid was tested by treating at different temperatures and time intervals, using qPCR.

for complete inactivation. Extending the reaction time to 5–30 min progressively reduced viral activity, yet infectious particles were still detectable after 30 min (Fig. 4B).

In contrast, 10% formalin and 4% paraformaldehyde showed complete inactivation. However, it is noteworthy that one of the four repeated experiments using 1% paraformaldehyde did not entirely inhibit viral activity (Table 4).

Our findings demonstrate that the Mpox virus is sensitive to heat treatment, with a minimum of 10 min at temperatures above 56°C, effectively inhibiting its activity. Commonly used laboratory lysis buffers and disinfectants also efficiently deactivated the virus at the recommended concentrations. However, the composition of RIPA buffer varies depending on its purpose and brand, resulting in varying outcomes. Additionally, paraformaldehyde at concentrations lower than 1% might leave viral residues. These two aspects will be further discussed.

## The Mpox virus exhibits high stability after multiple freeze-thaw cycles

Next, we evaluated the stability of infectious Mpox virus by subjecting it to freeze-thaw cycles. Virus aliquots were subjected to 1–6 freeze-thaw cycles. Plaque assays were used to determine the viral survival rate and to calculate the infectivity titer (Fig. 5A). Samples thawed once as a reference to calculate the relative percentage of viral activity after multiple freeze-thaw cycles. Interestingly, we observed that the viral titer increased during the first three freeze-thaw cycles, remained stable after the third cycle, and did not show any decrease in infectivity even at the end of the experiment (Fig. 5B). Furthermore, we examined whether freeze-thaw cycles degraded viral nucleic acids and affected qPCR results. While a slight decrease in the Ct value was observed after one or more freeze-thaw cycles, there was no significant difference across all six freeze-thaw cycle samples (Fig. 5C).

The Mpox virus exhibited remarkable stability during multiple freeze-thaw cycles, as evidenced by the absence of a reduction in viral titer following six cycles. Moreover, qPCR analysis revealed no observable impairment to viral nucleic acid integrity, indicating the resistance of the viral genome to damage under these conditions.

## DISCUSSION

Cell susceptibility is a critical determinant of viral infection, which depends on host cells for replication and propagation. The ability of a virus to infect a specific cell type is determined by the presence of specific cell-surface receptors that facilitate binding and entry, such as the sialic acid, which served as a crucial receptor for the Influenza virus (24, 25). Previous studies have elucidated the unique cellular entry mechanisms of the vaccinia virus, which may involve membrane fusion or endocytosis (26). It has been reported that the Mpox virus shares similar receptors, such as heparan sulfate,

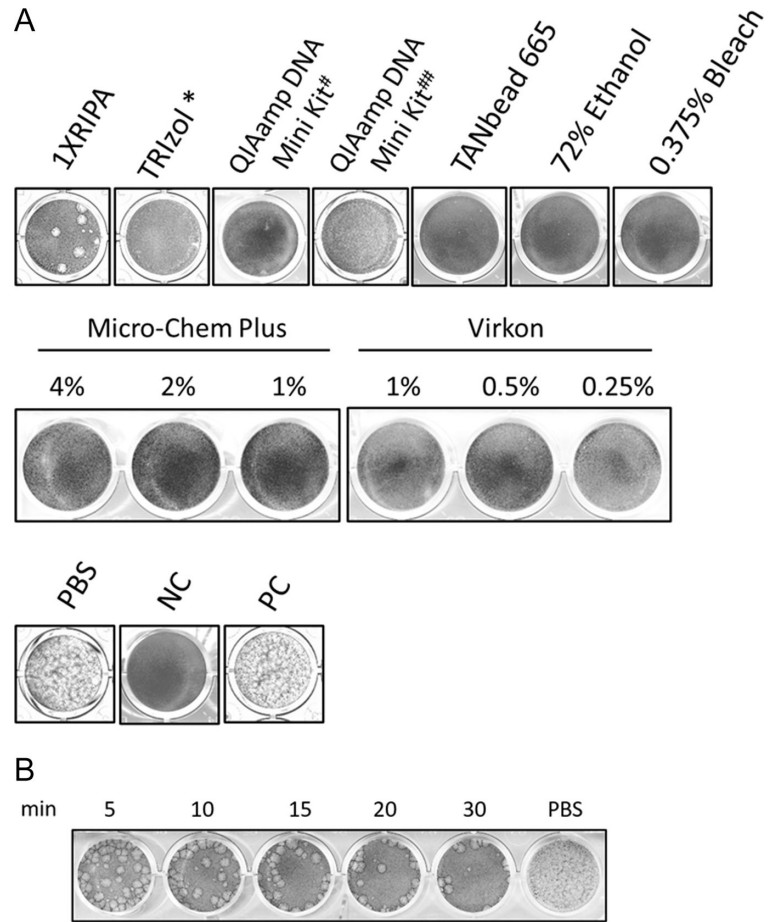

**FIG 4** The efficiency of inactivating the Mpox virus by various reagents. (A) Mpox virus was treated with various lysis buffers or disinfectants at a viral load of $8 \times 10^4$ PFU. The efficacy of inactivation was examined using a plaque formation assay. (B) The efficacy of RIPA buffer inhibiting Mpox virus at time intervals. (*: the volume ratio of liquid viral sample, and Trizol was 1:3; #: following the manual to stop at step 2; and ##: following the manual to stop at step 4).

chondroitin sulfate, and glycosaminoglycan (27), with the vaccinia virus. However, the precise receptors involved in these interactions remain unknown.

In the susceptibility study, we removed non-adsorbed viruses and evaluated the number of viruses attached to the cells as a baseline for calculating the virus replication rate. This approach better reflects the actual viral replication fold change than the total number of viruses added as a reference. The baseline viral load revealed that different cell types have different affinities for the Mpox virus. This suggests that cells with stronger binding affinities and their surface molecules could be further investigated to understand the diversity of virus receptors on host cells. Furthermore, we observed that the number of copies per milliliter in the viral growth curve and susceptibility test was similar for nearly all cell lines at 96 h post-infection (Fig. 1B). However, the growth curves of actual infectious particles revealed that only BSC-1 cells maintained high viral sensitivity, with a higher fold change in virus replication. Analysis of copies-to-PFU ratios revealed that only the BSC-1 and Vero E6 demonstrated better virus packaging efficiency, as evidenced by the average log numbers ranging from 0.74 to 1.16. In contrast, the remaining cell lines averaged between 1.98 and 2.19 (Table 5) (28, 29). In this section, the replication of the Mpox virus can be observed in multiple cell types. However, when generating virus stocks, the ratio of viral copies per PFU might be of critical considera-

**TABLE 3** The condition of lysis buffer and disinfections treated with Mpox virus

| Items | Reagent conc./volume (µL) | Input viral (PFU)/ volume (µL) | Final conc. | Supplier | Fully inactive rate |
|---|---|---|---|---|---|
| **Lysis buffer** | | | | | |
| RIPA | 2×/100 | $8 \times 10^4$/100 | 1X | Merck (Cat# 20-188) | 0% (0/8) |
| TRIzol | 1×/300 | $8 \times 10^4$/400 | 0.75X | Sigma (Cat# T9424) | 100% (8/8) |
| QIAamp DNA Mini (to step 2) | 1×/200 | $8 \times 10^4$/140 | 1X | QIAGEN (Cat# 51104) | 100% (8/8) |
| QIAamp DNA Mini (to step 4) | 1×/200 | $8 \times 10^4$/140 | 1X | QIAGEN (Cat# 51104) | 100% (8/8) |
| TANbead 665 | 1×/650 | $8 \times 10^4$/300 | 1X | TANbead (Cat# M665S46) | 100% (8/8) |
| **Disinfection** | | | | | |
| Ethanol | 80%/900 | $8 \times 10^4$/100 | 72% | Merck (Cat# 111727) | 100% (8/8) |
| Bleach | 0.375%/900 | $8 \times 10^4$/100 | 0.338% | Magic Amah | 100% (8/8) |
| Micro-Chem Plus | 4%/300 | $8 \times 10^4$/100 | 3% | NCL | 100% (8/8) |
| Micro-Chem Plus | 2%/300 | $8 \times 10^4$/100 | 1.5% | NCL | 100% (8/8) |
| Micro-Chem Plus | 1%/300 | $8 \times 10^4$/100 | 0.75% | NCL | 100% (8/8) |
| Virkon | 1%/900 | $8 \times 10^4$/100 | 0.9% | Du Pont | 100% (8/8) |
| Virkon | 0.5%/900 | $8 \times 10^4$/100 | 0.45% | Du Pont | 100% (8/8) |
| Virkon | 0.25%/900 | $8 \times 10^4$/100 | 0.225% | Du Pont | 100% (8/8) |

tion. Cells with a lower copies-to-PFU ratio, such as BSC-1, exhibit better viral replication and packaging efficiency, concurrently yielding fewer defective particles.

Several cell lines, such as BSC-1, MRC-5, VeroE6, and 293T, maintained similar viral titers between P1 and P2 (Table 1). This observation likely reflects their high permissiveness to Mpox virus, as these cells can efficiently support viral replication without the need for adaptation. The lack of further increase in P2 may indicate that virus replication had already reached a plateau. In contrast, BHK-21 and HEp-2 cells showed an increase in P2 titers. These cells initially supported lower replication in P1, but improved yields in P2 suggest that Mpox virus may have begun adapting to these less-permissive environments. Similar patterns of virus adaptation through serial passage have been reported in other poxviruses (30). Conversely, viral titers declined in P2 cultures of RD, Huh-7, MDCK, and CHO cells, implying that these cell lines are only semi-permissive for Mpox virus. The reduced yield is most likely due to an intracellular block rather than inefficient viral entry. For instance, previous studies demonstrated that in CHO cells, vaccinia virus and related orthopoxviruses abort at the stage of intermediate protein synthesis, resulting in markedly lower progeny production. A similar restriction mechanism may therefore explain the replication we observed in these four cell lines (31). These patterns are evident not only from the copies to PFU ratios in Table 1 but also from the CPE observed at 96 h post-infection in P1 and P2 cultures. In highly permissive cell lines such as BSC-1 and VeroE6, extensive CPE was already visible at P1-96h, and by P2-96h, the monolayers were nearly destroyed. In moderately permissive cell lines such as BHK-21 and HEp-2, the degree of CPE at 96 h increased from P1 to P2, suggesting progressive viral adaptation. In contrast, cell lines that likely restrict Mpox replication, including MDCK and CHO,

**TABLE 4** The efficiency of formalin or paraformaldehyde inactivity Mpox virus

| Treatment | | CPE (72 h post-infection) | | | | Fully inactive rate |
|---|---|---|---|---|---|---|
| Items | Input viral load | Replicate 1 | Replicate 2 | Replicate 3 | Replicate 4 | |
| **Formalin** | | | | | | |
| 10% | 0.1 MOI | ND[a] | ND | ND | ND | 100% (4/4) |
| 5% | 0.1 MOI | ND | ND | ND | ND | 100% (4/4) |
| 1% | 0.1 MOI | ND | ND | ND | ND | 100% (4/4) |
| **Paraformaldehyde** | | | | | | |
| 4% | 0.1 MOI | ND | ND | ND | ND | 100% (4/4) |
| 1% | 0.1 MOI | ND | ND | CPE | ND | 75% (3/4) |

[a]"ND" stands for non-detected, indicating that no cytopathic effect (CPE) was observed according to our experimental method.

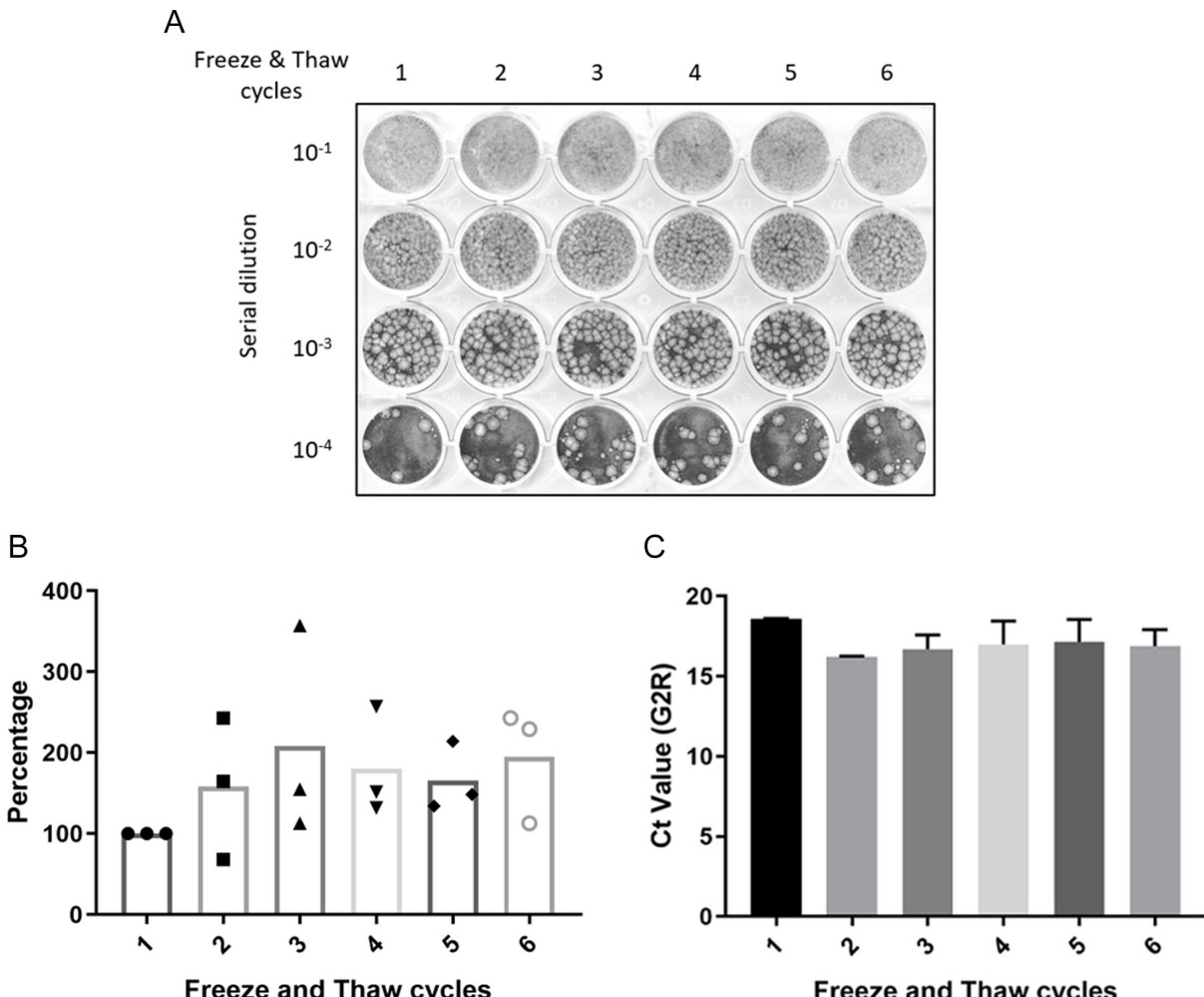

**FIG 5** The stability assessment of Mpox virus subjected to multiple freeze-thaw cycles. (A) The virus underwent repeated freeze-thaw cycles, and its viability was subsequently determined by plaque formation assay. (B) The viral activity was tested and calculated into percentages compared with the first cycle. (C) The stability of Mpox virus nucleic acid was tested after multiple freeze-thaw cycles, using qPCR.

showed little to no CPE throughout the experiment (Fig. S2). Together, these results highlight distinct patterns of host-cell interactions with Mpox virus, including stable support, adaptation, and restriction. These differences are important considerations when selecting cell lines for virus propagation or antiviral screening.

The plaque assay is a fundamental technique for determining the viral titers and remains the gold standard for accurately measuring infectious virus particles (32). Traditionally, agarose or methylcellulose has been used in solid or semisolid overlays to prevent viral spread through liquid convection. However, a simplified and safer approach is required in laboratories with high biosafety levels, such as BSL-3 facilities. Therefore,

**TABLE 5** The ratio of viral copies and infectious particles during the growth curve analysis

| | Copies/mL (Log) | | | | | PFU/mL (Log) | | | | | Copies/PFU ratio (Log) | | | | |
|---|---|---|---|---|---|---|---|---|---|---|---|---|---|---|---|
| | 6 h | 24 h | 48 h | 72 h | 96 h | 6 h | 24 h | 48 h | 72 h | 96 h | 6 h | 24 h | 48 h | 72 h | 96 h |
| BS-C-1 | 2.76 | 4.26 | 6.51 | 7.30 | 7.75 | 1.42 | 2.42 | 5.12 | 6.82 | 6.98 | 1.34 | 1.84 | 1.39 | 0.48 | 0.77 |
| Vero E6 | 1.83 | 4.37 | 5.15 | 5.57 | 5.70 | 1.42 | 3.03 | 4.64 | 5.02 | 4.80 | 0.41 | 1.34 | 0.51 | 0.55 | 0.90 |
| MRC-5 | 4.14 | 6.36 | 7.64 | 7.27 | 6.56 | 2.42 | 4.25 | 5.31 | 5.14 | 4.54 | 1.72 | 2.10 | 2.33 | 2.13 | 2.02 |
| RD | 4.48 | 5.97 | 7.35 | 7.21 | 6.54 | 2.28 | 4.20 | 5.27 | 4.30 | 4.61 | 2.19 | 1.77 | 2.08 | 2.91 | 1.93 |
| BHK-21 | 5.22 | 5.18 | 7.48 | 7.13 | 7.50 | 2.29 | 4.68 | 5.50 | 4.99 | 4.13 | 2.93 | 0.50 | 1.98 | 2.14 | 3.37 |
| MM55.K | 5.01 | 4.48 | 6.16 | 6.54 | 6.07 | 2.62 | 3.24 | 3.92 | 4.28 | 4.30 | 2.39 | 1.24 | 2.24 | 2.26 | 1.77 |

agarose was phased out as an overlay medium for the plaque assays. We compared the conditions and effectiveness of methylcellulose and microcrystalline cellulose as overlay media in plaque assays. Mpox virus is predominantly transmitted via cell-to-cell contact. The plaque assay using the E-2 medium alone may generate numerous comet-shaped plaques, which are smaller and can lead to misinterpretation of viral titer calculations. Furthermore, the virus titers calculated from the E-2 medium may be overestimated. Nevertheless, this approach can be used if the plates are incubated for only 2–3 days, given the significant size difference between the comet-shaped and primary virus plaques. Adding at least 0.4% methylcellulose or 0.63% microcrystalline cellulose effectively mitigated the appearance of comet-shaped plaques in plaque assays.

Due to the absence of previous records of Mpox cases in Taiwan, no live virus was available to test the efficacy of various inactivation conditions. As a result, inactivation standards for the Mpox virus were previously established based on literature references and data from related viruses, such as the vaccinia virus (33). This study investigated the inactivation conditions and stability of Mpox virus from four different perspectives to provide a foundation for future guidelines in Taiwan.

The first aspect explored the inhibitory effect of heat treatment on the Mpox virus, which revealed 10 min at 56°C effectively inactivated the virus. Our findings were largely consistent with those of Kaplan et al. and Christophe et al., although the latter study applied 56°C for 30 min, which was insufficient for complete inactivation. However, the main difference between our study and Christophe's study lies in the viral load used for the reaction, with Christophe's study using approximately $7 \times 10^6$ PFU of the virus. In contrast, we used $8 \times 10^4$ PFU of the virus. Additionally, they used a water bath for heating, whereas we used a more precise PCR machine for temperature control, which may account for the inconsistencies in the data between the two studies (34, 35). However, the outcomes of both studies indicate the Mpox virus to be thermosensitive. We observed that virus inactivation at 56°C resulted in slightly higher Ct values than at higher temperatures. This indicated that nucleic acid-based detection following 56°C treatment may yield lower viral genome or copy numbers than treatments at higher temperatures. We hypothesize that temperatures below 65°C may preserve nuclease activities, leading to degradation of the viral genome despite viral inactivation (36, 37). These findings highlight the importance of considering this phenomenon in future nucleic acid-based viral detection strategies.

Second, understanding the efficacy of various lysis buffers and disinfectants in suppressing Mpox virus is essential for effective pathogen control in public health settings and for enhancing laboratory safety practices. Our testing of various reagents could completely inhibit the Mpox virus at standard working concentrations, consistent with results reported in other publications (38–40). RIPA buffer is a widely used reagent for protein extraction and an essential component of many protein experiments. Therefore, verifying the conditions under which the RIPA buffer can effectively inactivate the Mpox virus is critical for future protein-related experiments. Surprisingly, our findings indicated that even after 10 min of incubation with the Mpox virus at the standard working concentration, viral plaques were formed despite a 99.94% inhibition rate (Fig. 4A; Table 3). Subsequently, we conducted a time-course experiment with RIPA buffer and discovered that complete viral inactivation was not achieved even after 30 min of exposure (Fig. 4B). After examining the components of the commercial RIPA buffer used in this study, we found that it lacked 0.1% SDS, unlike the conventional recipe (41). We then procured a different brand of RIPA buffer containing 0.1% SDS and supplemented the original RIPA buffer used in this experiment with 0.1% SDS. The virus inhibition experiment was repeated, and the results demonstrated complete viral inactivation (data not shown).

Third, we investigated the inhibitory effects of commonly used fixatives on the Mpox virus. Formalin or paraformaldehyde was widely used as a fixative for immunofluorescence staining, formalin-fixed paraffin embedding, and *in situ* hybridization (42). The effectiveness of several concentrations commonly used for cell fixation in suppressing

the activity of the Mpox virus was evaluated in our study. Our results revealed that standard working concentrations of formalin effectively inhibited viral activity for over 30 min. Notably, incomplete inactivation was observed in one out of four replicates with 1% paraformaldehyde, as evidenced by CPE observations (Table 3). These findings are consistent with a previous study by Ellen et al., which reported that 1.5% paraformaldehyde treatment for 30 min could suppress approximately $10^7$ PFU of the vaccinia virus, whereas 0.3% can suppress $10^2$ PFU of the virus (43). Our study used a virus concentration approaching $10^5$ PFU, which may explain the observed CPE under 1% paraformaldehyde conditions, consistent with similar studies despite the differences between viruses.

During freezing, the formation of ice crystals within cells can puncture the cell membrane (44). This phenomenon can also occur in enveloped viruses such as influenza, whose outer layer consists of a phospholipid bilayer (45). Although the Mpox virus is also enveloped, its viral genome may be damaged upon exposure to multiple freeze-thaw cycles (46). Therefore, testing the tolerance of the Mpox virus to freeze-thaw cycles is highly significant for specimen preservation, transportation, and experimental operations. Our results demonstrate that Mpox virus exhibits a considerable degree of tolerance to repeated freeze-thaw cycles, which may be related to its life cycle. Most virus particles accumulate inside cells, where the microenvironment during freezing provides a relatively high protein concentration that serves as a protective mechanism.

In this study, several limitations should be considered. First, using a single virus strain of virus limits the generalizability of our findings. Mpox virus includes clade I and clades IIa and IIb. Due to the absence of local cases in Taiwan, only the IIb lineage of the virus was available for validation. Additionally, a comparative analysis with other viruses or orthopoxviruses was not performed. Previous literature demonstrates extensive research on the vaccinia virus, thus guiding our primary focus toward studying the Mpox virus. Second, the range of cell lines used in this study was relatively limited. The chosen cell lines primarily represent those commonly employed in laboratory settings and do not comprehensively simulate infection across diverse organ-derived cell lines. To address this limitation, we have initiated plans to expand our cell repository to facilitate broader susceptibility testing.

In conclusion, our study provides evidence of the susceptibility of several widely used cell lines to Mpox virus. These findings have important implications for future research, particularly in selecting appropriate expression systems. Moreover, we validated the efficacy of inactivation protocols for the Mpox virus using a range of disinfectants and lysis reagents commonly employed in laboratory and environmental disinfection. The insights obtained from this research have potential applications in experimental design, laboratory sterilization, and environmental sanitation practices.

## ACKNOWLEDGMENTS

We sincerely thank the management team of the BSL-4 laboratory at our research institute for their invaluable support in laboratory scheduling, maintenance, and equipment provision.

The research funding for this study was primarily provided by the Institute of Preventive Medicine at the National Defense Medical University, Taiwan (No. 112-G2-2), with partial support from research grants from the Centers for Disease Control, Taiwan (No. 112M0021). We are also deeply grateful to the National Science and Technology Council, Taiwan (NSTC 114-2321-B-016-004), for supporting this research.

The authors declare no known competing financial interests or personal relationships.

## AUTHOR AFFILIATIONS

[1]Graduate Institute of Medical Sciences, National Defense Medical University, Taipei, Taiwan

[2]Graduate Institute of Biodefense, National Defense Medical University, Taipei, Taiwan

[3]Institute of Preventive Medicine, National Defense Medical University, Taipei, Taiwan

[4]Department of Physiology and Biophysics, Graduate Institute of Physiology, National Defense Medical University, Taipei, Taiwan

[5]Division of Infectious Diseases and Tropical Medicine, Department of Internal Medicine, Tri-Service General Hospital, National Defense Medical University, Taipei, Taiwan

[6]Center for Diagnostics and Vaccine Development, Centers for Disease Control, Taipei, Taiwan

[7]Graduate Institute of Applied Science and Technology, National Taiwan University of Science and Technology, Taipei, Taiwan

[8]Department of Cosmetic Science, Vanung University, Taoyuan City, Taiwan

## AUTHOR ORCIDs

Shu-Chen Hsu http://orcid.org/0000-0002-5097-2540
Tein-Yao Chang http://orcid.org/0000-0002-7731-8493

## AUTHOR CONTRIBUTIONS

Shu-Chen Hsu, Conceptualization, Data curation, Formal analysis, Methodology, Visualization, Writing – original draft | Ping-Cheng Liu, Data curation, Methodology | Shan-Ko Tsai, Data curation, Methodology | An-Yu Chen, Data curation, Methodology | Hui-Ping Tsai, Supervision | Jun-Ren Sun, Supervision | Ti-Yu Li, Methodology | Pei-Yu Hsieh, Methodology | Jyh-Yuan Yang, Supervision | Tein-Yao Chang, Conceptualization, Formal analysis, Funding acquisition, Methodology, Project administration, Writing – review and editing

## ADDITIONAL FILES

The following material is available online.

### Supplemental Material

**Supplemental figures (Spectrum00803-25-s0001.docx).** Figures S1 and S2.

### Open Peer Review

**PEER REVIEW HISTORY (review-history.pdf).** An accounting of the reviewer comments and feedback.

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
