## [Reviewer comments · Microbiology Spectrum]

Microbiology Spectrum

Evaluating the Susceptibility of Various Common Cell Lines and Assessing Inactivation Conditions to Mpox Virus

Shu-Chen Hsu, Ping-Cheng Liu, Shan-Ko Tsai, An-Yu Chen, Hui-Ping Tsai, Jun-Ren Sun, Ti-Yu Li, Pei-Yu Hsieh, Jyh-Yuan Yang, and Tein-Yao Chang

Corresponding Author(s): Tein-Yao Chang, National Defense Medical Center

Review Timeline:

Submission Date:	March 27, 2025
Editorial Decision:	May 29, 2025
Revision Received:	July 10, 2025
Accepted:	August 26, 2025

Editor: Justin Jang Hann Chu

Reviewer(s): Disclosure of reviewer identity is with reference to reviewer comments included in decision letter(s). The following individuals involved in review of your submission have agreed to reveal their identity: Chee-Keng Mok (Reviewer #1)

Transaction Report:

DOI: <https://doi.org/10.1128/spectrum.00803-25>

Re: Spectrum00803-25 (Evaluating the Susceptibility of Various Common Cell Lines and Assessing Inactivation Conditions to Mpox Virus)

Dear Dr. Tein-Yao Chang:

Thank you for the privilege of reviewing your work. Below you will find my comments, instructions from the Spectrum editorial office, and the reviewer comments.

Revision Guidelines

Sincerely,
Justin Jang Hann Chu
Editor
Microbiology Spectrum

Reviewer #1 (Comments for the Author):

Shu-Chen Hsu and her team have evaluated and verified specific conditions for Mpox virus isolation and inactivation. These are critical findings for mitigating biorisks in downstream studies.

Please find my comments below:

1. Statistical analysis should be performed for all experiments. The number of replicates or repeats should be clearly stated to ensure the reproducibility of the results.
2. In the susceptibility study using multiple cell lines, P2 was inoculated from P1, and no significant fold change in replication was observed compared to P1/P0. This observation should be discussed further, including (but not limited to) the level of cell death or cytopathic effect (CPE) at 4 days post-infection (dpi) for both P1 and P2. A similar observation was seen in Fig. 1B, where virus titers dropped at 96 hours post-infection (hpi). Could this be due to cell death?
3. Data presentation in Table 2 should be improved. Please include the units, volumes, and any specific conditions used.
4. Line 234-235 should be revised for clarity-it is currently difficult to understand. Also, the statement in line 251 regarding "99.99% inhibition by RIPA" lacks supporting information and should be substantiated or rephrased.
5. Inactivation method validation is a serious concern in the biosafety ecosystem. It would be helpful to provide detailed information on raw materials and conditions used, such as the type of heat equipment, tubes or containers, incubation time, temperature, and whether validation was done immediately after treatment or after storage (e.g., in a freezer).
6. In the section on inactivation using lysis buffer or disinfectant, filters were used to remove chemicals. Please clarify whether the filter is chemically resistant and whether its integrity is preserved to avoid accidental passage of virus particles. Additionally, what percentage of the resuspension was validated, as the buffer was reportedly resuspended in 4 mL?
7. The experimental design for the fixatives is unclear. Were the viruses fixed, filtered, and then validated? What were the fixation conditions (e.g., duration, temperature, initial virus load)? If viruses were fixed together with infected cells, how was it ensured that no chemical residues remained in the lysate or cells before inoculating new cell lines for validation?

Reviewer #3 (Comments for the Author):

The following are the comments on the manuscript titled: "Evaluating the Susceptibility of Various Common Cell Lines and Assessing Inactivation Conditions to Mpox Virus".

Overall, the manuscript is well-written, and the methods used to study virus inactivation is widely used and well-accepted internationally. However, besides presenting the inactivation methods and their efficiency in inactivating the Mpox virus, the paper could be strengthened by the addition of a product neutralization and cytotoxicity assessment to show that the disinfectants used in the study, when used at the concentrations that they were tested in, did not cause any cytotoxic effects to the cells.

In addition, there are a number of spelling errors that should be corrected:

Line 50: The word 'Poxviridae' should be italicised.

Line 60: Remove the comma after the word 'transmission'.

Figure 4A: Typo for 'Micro-Chen plus'. In addition, why is there one lone plaque in the figure for TANbead665, 75% ethanol and 0.375% Bleach (seen in Figure 4A) when table 3 states that the fully inactive rate is 100%? There should be no plaques if it is 100%. These plaques also seem to all eerily occur on the same zone in Figure 4A. Also, what is the exact ethanol concentration used? Figure 4A states 75%, but table 3 states 80%.

Table 3: Again, typo for 'Micro-Cham plus'.

Reviewer #1 (Comments for the Author):

1. Statistical analysis should be performed for all experiments. The number of replicates or repeats should be clearly stated to ensure the reproducibility of the results.

Thank you for highlighting the need for clearer statistical reporting. We have revised the Materials and Methods section to specify the exact number of replicates for each experiment.

Additionally, we have added a new subsection (2.10 Statistical Analysis) that describes our data processing workflow in detail. All experiments with quantitative comparisons now include appropriate statistical tests, and the results section has been revised to ensure that these analyses are clearly reported to enhance transparency and reproducibility.

2. In the susceptibility study using multiple cell lines, P2 was inoculated from P1, and no significant fold change in replication was observed compared to P1/P0. This observation should be discussed further, including (but not limited to) the level of cell death or cytopathic effect (CPE) at 4 days post-infection (dpi) for both P1 and P2. A similar observation was seen in Fig. 1B, where virus titers dropped at 96 hours post-infection (hpi). Could this be due to cell death?

Thank you for your professional and insightful comment. In response, we have added a new paragraph to the Discussion section to address your observation regarding the limited fold change in viral replication between P1 and P2 in certain cell lines.

The added discussion reads as follows:

“Several cell lines, such as BSC-1, MRC-5, VeroE6 and 293T, maintained similar viral titers between P1 and P2 (Table 1). This observation likely reflects their high permissiveness to Mpox virus, as these cells can efficiently support viral replication without the need for adaptation. The lack of further increase in P2 may indicate that virus replication had already reached a plateau. In contrast, BHK-21 and HEp-2 cells showed an increase in P2 titers. These cells initially supported lower replication in P1, but improved yields in P2 suggest that Mpox virus may have begun adapting to these less-permissive environments. Similar patterns of virus adaptation through serial passage have been reported in other poxviruses³⁰. Conversely, viral titers declined in P2 cultures of RD, Huh-7, MDCK, and CHO cells, implying that these cell lines are only semi-permissive for Mpox virus. The reduced yield is most likely due to an intracellular block rather than inefficient viral entry. For instance, previous studies demonstrated that in CHO cells, vaccinia virus and related orthopoxviruses abort at the stage of intermediate protein synthesis, resulting in markedly lower progeny production. A similar restriction mechanism may therefore explain the replication we observed in these four cell lines³¹. These patterns are evident not only from the copies to PFU ratios in Table 1 but also from the CPE observed at 96 h post-infection in P1 and P2 cultures. In highly permissive cell lines such as BSC-1 and VeroE6, extensive CPE was already visible at P1-96h, and by P2-96h, the monolayers were nearly destroyed. In moderately permissive cell lines such as BHK-21 and HEp-2, the degree of CPE at 96 h increased from P1 to P2, suggesting progressive viral adaptation. In contrast, cell lines that likely restrict Mpox replication, including MDCK and CHO, showed little to no CPE throughout the experiment

(Supplementary Fig. S2). Together, these results highlight distinct patterns of host-cell interactions with Mpx virus, including stable support, adaptation, and restriction. These differences are important considerations when selecting cell lines for virus propagation or antiviral screening.”

3. Data presentation in Table 2 should be improved. Please include the units, volumes, and any specific conditions used.

Thank you for your suggestion. We have revised Table 2 to include the units, volumes, and any relevant experimental conditions to improve clarity and ensure all necessary details are provided

4. Line 234-235 should be revised for clarity-it is currently difficult to understand.

Thank you very much for pointing out the unclear wording. We have revised lines 234–235 for improved clarity. The revised sentence now reads:

" Our findings indicate that heat treatment of the Mpx virus at 56 °C for 5 minutes can reduce plaque counts while extending the treatment to 10 minutes or longer achieves complete viral inactivation (Fig 3A)”

5. Also, the statement in line 251 regarding "99.99% inhibition by RIPA" lacks supporting information and should be substantiated or rephrased.

Thank you for your valuable feedback. We fully appreciate your observation regarding the statement in line 251 concerning the "99.99% inhibition by RIPA." The calculation of the 99.99% inhibition rate, as mentioned in our manuscript, is based on the experimental setup in which we consistently used 8×10^4 pfu of Mpx virus (Table 3). As shown in Figure 4A, the RIPA group still showed 7 plaques, leading to the calculation of the viral inhibition rate as follows: $(1 - (7/80000)) * 100\% = 99.99\%$.

Upon further review, although we did include a positive control (PC) to validate our experimental procedure, we did not conduct quantification of the PC. Thus, the accurate inhibition rate should be calculated based on the actual pfu of the PC. We now recognize that the originally reported 99.99% inhibition rate was not entirely correct. To avoid any potential misinterpretation, we have revised the statement in line 251 accordingly. The revised text is as follows:

"Unexpectedly, a few viral plaques persisted after repeated treatments with commercial RIPA buffer, indicating that a 10-min exposure was insufficient for complete inactivation. Extending the reaction time to 5–30 min progressively reduced viral activity, yet infectious particles were still detectable after 30 min (Fig. 4B).”

We sincerely thank you for your thorough review, which has helped improve the accuracy and clarity of our manuscript.

6. Inactivation method validation is a serious concern in the biosafety ecosystem. It would be helpful to provide detailed information on raw materials and conditions used, such as the type of heat equipment, tubes or containers, incubation time, temperature, and whether validation was done

immediately after treatment or after storage (e.g., in a freezer).

We fully agree that detailed reporting of materials and conditions is essential for biosafety validation and for enabling other researchers or biosafety practitioners to reproduce our work. After carefully reviewing our Materials and Methods section, we realized that several procedural details had not been described with sufficient precision. We have therefore revised the manuscript to include the following information:

- Heat-treatment equipment – heat treatments were performed in a PCR thermocycler (TCLT9610, Blue-Ray Biotech, Taiwan), which provides rapid temperature equilibration and precise control.
- Reaction vessels – Virus aliquots were placed in 0.2 mL polypropylene PCR strip tubes (Cat# MB-P08-A, Gunster, Taiwan).
- Incubation parameters – Treatment temperatures (56 °C, 65 °C, 95 °C) and exposure times (1, 5, 10, 15 and 30 min).
- Post-treatment handling – Viral inactivation was validated immediately after heat treatment for all assays. In the separate freeze–thaw stability study, aliquots were stored at –80 °C between cycles; validation was carried out immediately after the final cycle.
- Plasticware and reagents – Catalogue numbers and manufacturers for all tubes, plates, and media are now provided to ensure complete traceability.

We sincerely appreciate your guidance in strengthening the manuscript.

7. In the section on inactivation using lysis buffer or disinfectant, filters were used to remove chemicals. Please clarify whether the filter is chemically resistant and whether its integrity is preserved to avoid accidental passage of virus particles. Additionally, what percentage of the resuspension was validated, as the buffer was reportedly resuspended in 4 mL?

Thank you for your insightful question. We appreciate the opportunity to clarify these important details regarding filter compatibility, integrity, and the resuspension volume used in our assays.

1. Chemical resistance and filter integrity

- In the Amicon centrifugal filter User Guide, the chemical compatibility of the filter is detailed. Based on the safety data sheets of the commercial disinfectants and lysis buffers used in our study, we calculated the concentrations of chemicals in the formulations. For the most part, these formulations fall within the chemical compatibility guidelines of the Amicon centrifugal filter, except for Trizol.
- According to our formulation, during the first wash process, the filter was exposed to approximately 3.75–5% of phenol. The User Guide recommends that the concentration of phenol should be $\leq 1\%$, and if higher concentrations are used, solvent blanks should be included.
- To confirm that the filter retained its integrity and functionality under our experimental conditions, we conducted an additional validation assay. Briefly, 300 μL of Trizol was diluted with PBS to 4 mL and centrifuged at 3,500 rpm until $\sim 250 \mu\text{L}$ remained. PBS was then added to restore the volume to 4 mL, and the centrifugation step was repeated twice more to simulate the washing procedure. In parallel, a control filter was processed with PBS alone. All conditions were tested in duplicate. Subsequently, 1 mL of vaccinia virus (1×10^6 pfu) was applied to either

a control filter or the Trizol-treated filter and centrifuged at 3,500 rpm, concentrating the retentate to 250 μ L. We collected both the retentate above the filter and the pass-through filtrate below it to confirm that no virus had leaked into the filtrate due to filter failure. Viral titers were determined by plaque assay and compared between the treated and untreated filters. Because work with the Mpxv virus in a BSL-4 laboratory requires lengthy biosafety approvals, we used VACV, another member of the *Poxviridae* family, as a surrogate within the limited time available for our revision. The experiment was designed to demonstrate that the Trizol concentration applied under our conditions does not compromise filter integrity or performance. Moreover, previous studies indicate that VACV closely resembles Mpxv virus in both size and structure, further supporting its suitability for this purpose.

- (A) shows that filters pre-treated with either Trizol or PBS concentrate vaccinia virus with comparable efficiency. The number of plaques in the retentate is essentially identical for both treatments. Quantification of the flow-through fractions further confirms filter integrity, as no detectable virus leaked into the filtrate after either treatment. Bargraph analysis (B) of duplicates yields mean titers of $10^{6.63 \pm 0.11}$ pfu for the Trizol group and $10^{6.66 \pm 0.02}$ pfu for the PBS control, indicating no significant difference. Collectively, these results demonstrate that our disinfectant evaluation protocol is workable, under our conditions tested, Trizol does not compromise filter performance.

2. Revision to the Methods section:

We have revised the relevant section in the Materials and Methods for clarity, as follows:

“For evaluating the effect of lysis buffers and disinfectants, 8×10^4 PFU of Mpxv virus was mixed with the respective solutions listed in Table 3 and incubated for 10 min at room temperature. The reaction volume was then adjusted to 4 mL with phosphate-buffered saline (PBS) and passed through an Amicon Ultra-4 100 kDa centrifugal filter (Millipore, Ireland) at 3,500 rpm by centrifuge (DSC-N158A, Digisystem Laboratory Instrument, Taiwan) until $\sim 250 \mu$ L remained. The retentate was brought back to 4 mL with PBS, and the wash was repeated twice more to remove residual chemicals. The retentate remaining on the filter was brought to a final volume of 500 μ L with E-2 medium. Each experimental condition was tested in quadruplicate. Then, the viruses were inoculated into BSC-1 cells in a 24-well plate and incubated for three days. The cells were fixed and stained with crystal violet to visualize the viral plaques.”

8. The experimental design for the fixatives is unclear. Were the viruses fixed, filtered, and then validated? What were the fixation conditions (e.g., duration, temperature, initial virus load)? If viruses were fixed together with infected cells, how was it ensured that no chemical residues remained in the lysate or cells before inoculating new cell lines for validation?

Thank you very much for your valuable suggestion. We fully acknowledge and have considered the potential impact of formalin residue on the experiment.

1. Experimental design and rationale

- We followed the general approach of Alexander S. Jureka *et al.* (Viruses, 2020; 12(6):622), where infected monolayers are fixed *in situ* to model inactivation procedures for cell-associated virus.
- In our assay, BSC-1 cells were infected with Mpox virus at an MOI of 0.1 and incubated for 24 hours to allow intracellular replication.

2. Fixation conditions

- After infection, the medium was removed, and 1 mL of the chosen fixative (10%, 5%, or 1% formalin, or 4% or 1% paraformaldehyde) was added per well.
- Fixation was performed for 1 hour at room temperature.
- Mock-infected controls underwent the same fixation procedures.

3. Ensuring removal of fixative residues

- After the 1-hour fixation, fixative solutions were discarded, and monolayers were washed three times with 1 mL PBS per wash to remove residual chemicals.
- This multi-wash step is critical to ensure that no active fixative remained before subsequent procedures.

4. Revision to the Materials and Methods:

We have made the following revisions to the description:

“To validate the effect of formalin and paraformaldehyde on inactivating the virus. BSC-1 cells were grown to full confluence in 24-well plates. After PBS washes twice, the cells were infected with the virus at an MOI of 0.1 and incubated for 24 hours. The medium was then removed, and 1 mL of a chosen fixative was added to each well: 10 %, 5 %, or 1 % formalin, or 4 % or 1 % paraformaldehyde. Mock-infected controls were subjected to the same fixation procedures in parallel. Viral-control wells received 1 mL PBS instead of fixative. All conditions were set up in four replicates. Fixation lasted 1 hour at room temperature. The fixatives were discarded, and the monolayers were rinsed three times with 1 mL PBS to remove chemical residues. Each well was then loaded with 500 μ L of PBS, and the plate underwent three freeze–thaw cycles to release virus from the cells. The whole suspension was collected, and 200 μ L of it was used to infect fresh confluent BSC-1 monolayers. After 1 hour of adsorption, the inoculum was removed, 0.5 mL E-2 medium was added, and the cultures were incubated for 72 hours. Finally, cells were fixed with 10 % formalin for 1 hour and stained with crystal violet to visualize the plaques.”

Reviewer #3 (Comments for the Author):

1. Overall, the manuscript is well-written, and the methods used to study virus inactivation is widely used and well-accepted internationally. However, besides presenting the inactivation methods and their efficiency in inactivating the Mpox virus, the paper could be strengthened by the addition of a product neutralization and cytotoxicity assessment to show that the disinfectants used in the study, when used at the concentrations that they were tested in, did not cause any cytotoxic effects to the cells.

Thank you for your valuable suggestion. We completely agree that assessing the potential cytotoxicity of disinfectants is important when evaluating virus inactivation efficacy. In our study, all disinfectants were used strictly at their standard working concentrations, as specified by the manufacturers. These are commercially approved products that have undergone regulatory safety testing and come with detailed technical documentation regarding safe use, including cytotoxicity data for general applications.

However, we acknowledge that performing a specific neutralization and cytotoxicity assessment under our experimental conditions would further strengthen the study by confirming that observed reductions in infectivity were not influenced by residual cytotoxic effects on the cell lines used. Although this was outside the scope of our current work, we appreciate the reviewer's point and will consider incorporating such assessments in our future research to improve the rigor and completeness of the methodology.

2. Line 50: The word 'Poxviridae' should be italicised.

Thank you for your kind reminder. I have made the requested revision and italicised the word "Poxviridae" in line 50, as suggested.

3. Line 60: Remove the comma after the word 'transmission'.

Thank you for your kind reminder. I have revised line 60 to remove the comma after the word "transmission", as suggested.

4. Figure 4A: Typo for 'Micro-Chen plus'.

Thank you for pointing that out. I have corrected the spelling in Figure 4A to ensure accuracy. I appreciate your careful review.

5. In addition, why is there one lone plaque in the figure for TANbead665, 75% ethanol and 0.375% Bleach (seen in Figure 4A) when table 3 states that the fully inactive rate is 100%? There should be no plaques if it is 100%. These plaques also seem to all eerily occur on the same zone in Figure 4A.

Thank you for your careful observation. We appreciate the opportunity to clarify.

These marks in Figure 4A are **not viral plaques**, but artifacts caused by the pipette tip scraping the cell monolayer during liquid removal. Upon closer inspection, the edges of these marks are sharp, which is distinctly different from viral plaques, which typically have blurry edges.

This occurred in the BSL-4 isolator environment, where maneuverability is limited, and two operators worked simultaneously. Differences in technique led to consistent scraping marks, particularly on the right side of the wells. In addition, we conducted a repeat experiment with another person, and this issue did not occur. To avoid any confusion, we will replace the image in Figure 4A with the result from the repeat experiment. We appreciate your professional and rigorous approach to the presentation of our experimental results.

A, it can be seen that the experiment was performed by the same person, and there are marks from the pipette tip on the right side of each well. **B**, shows the scratches left by the tip (marked with red boxes) and the viral plaques (marked with black boxes). The edges of the scratches and plaques are clearly distinguishable. **C**, shows the results from the same experiment, but conducted by another, more experienced person, and there are no marks from the pipette tip.

6. Also, what is the exact ethanol concentration used? Figure 4A states 75%, but table 3 states 80%.

Thank you for your careful observation and for pointing out this inconsistency. The actual final ethanol concentration used in our experiments was 72%, as correctly indicated in Table 3.

- Specifically, we prepared 80% ethanol stock solution, and for each reaction, we mixed 900 μ L of this ethanol with 100 μ L of virus solution, resulting in a final concentration of 72% (calculated as $80\% \times (900 / (900 + 100))$).
- The label in Figure 4A incorrectly states 75%, which was based on our initial design plan. However, according to the standard preparation formula (as shown in Table 3), we consistently prepared 72% ethanol for the assays.
- This choice was deliberate to ensure that the final working concentration remained within the effective range of 70–75% commonly used for disinfection.

We acknowledge this labeling error in Figure 4A and will correct it to accurately reflect the 72% final concentration. Thank you very much for your helpful guidance in improving the clarity and consistency of our manuscript.

7. Table 3: Again, typo for 'Micro-Cham plus'.

Thank you for pointing that out. I have corrected the spelling in Table 3. I appreciate your careful review.

Re: Spectrum00803-25R1 (Evaluating the Susceptibility of Various Common Cell Lines and Assessing Inactivation Conditions to Mpox Virus)

Dear Dr. Tein-Yao Chang:

Your manuscript has been accepted, and I am forwarding it to the ASM production staff for publication. Your paper will first be checked to make sure all elements meet the technical requirements. ASM staff will contact you if anything needs to be revised before copyediting and production can begin. Otherwise, you will be notified when your proofs are ready to be viewed.

Sincerely,
Justin Jang Hann Chu
Editor
Microbiology Spectrum

Reviewer #1 (Comments for the Author):

All my comments have been addressed. For non-critical assays like the growth kinetic study, the use of duplicate wells provides preliminary data, but robust scientific validation would require at least two independent experiments (beyond technical replicates). Incorporating this in future work would further strengthen reproducibility and confidence in the findings.

Reviewer #3 (Comments for the Author):

The author managed to improve on their figures and phrasing of their manuscript.